# Neutralizing Antibodies in COVID-19 Patients and Vaccine Recipients after Two Doses of BNT162b2

**DOI:** 10.3390/v13071364

**Published:** 2021-07-14

**Authors:** Julien Favresse, Constant Gillot, Laura Di Chiaro, Christine Eucher, Marc Elsen, Sandrine Van Eeckhoudt, Clara David, Laure Morimont, Jean-Michel Dogné, Jonathan Douxfils

**Affiliations:** 1Department of Laboratory Medicine, Clinique St-Luc Bouge, 5004 Namur, Belgium; j.favresse@labstluc.be (J.F.); la188727@student.helha.be (L.D.C.); christine.eucher@slbo.be (C.E.); marc.elsen@slbo.be (M.E.); 2Department of Pharmacy, Namur Research Institute for Life Sciences, University of Namur, 5000 Namur, Belgium; constant.gillot@unamur.be (C.G.); jean-michel.dogne@unamur.be (J.-M.D.); 3Department of Internal Medicine, Clinique St-Luc Bouge, 5004 Namur, Belgium; sandrine.vaneeckhoudt@slbo.be; 4Qualiblood s.a., 5000 Namur, Belgium; clara.david@qualiblood.eu (C.D.); laure.morimont@qualiblood.eu (L.M.)

**Keywords:** COVID-19, SARS-CoV-2, neutralizing antibodies, humoral response, long-term kinetics

## Abstract

The evaluation of the neutralizing capacity of anti-SARS-CoV-2 antibodies is important because they represent real protective immunity. In this study we aimed to measure and compare the neutralizing antibodies (NAbs) in COVID-19 patients and in vaccinated individuals. One-hundred and fifty long-term samples from 75 COVID-19 patients were analyzed with a surrogate virus neutralization test (sVNT) and compared to six different SARS-CoV-2 serology assays. The agreement between the sVNT and pseudovirus VNT (pVNT) results was found to be excellent (i.e., 97.2%). The NAb response was also assessed in 90 individuals who had received the complete dose regimen of BNT162b2. In COVID-19 patients, a stronger response was observed in moderate–severe versus mild patients (*p*-value = 0.0006). A slow decay in NAbs was noted in samples for up to 300 days after diagnosis, especially in moderate–severe patients (r = −0.35, *p*-value = 0.03). In the vaccinated population, 83.3% of COVID-19-naive individuals had positive NAbs 14 days after the first dose and all were positive 7 days after the second dose, i.e., at day 28. In previously infected individuals, all were already positive for NAbs at day 14. At each time point, a stronger response was observed for previously infected individuals (*p*-value < 0.05). The NAb response remained stable for up to 56 days in all participants. Vaccinated participants had significantly higher NAb titers compared to COVID patients. In previously infected vaccine recipients, one dose might be sufficient to generate sufficient neutralizing antibodies.

## 1. Introduction

The revelation of SARS-CoV-2 RNA through a real-time reverse transcription polymerase chain reaction (RT-PCR) from nasopharyngeal swab samples is considered the gold standard method for the diagnosis of acute SARS-CoV-2 infection. Nevertheless, individuals with positive RT-PCR results represent only a limited fraction of all infections, given the limited availability and the brief time window in which RT-PCR testing presents the highest sensitivity [1,2].

The detection of specific antibodies following SARS-CoV-2 infection allows for the evaluation of the seroprevalence, the identification of convalescent plasma donors, the monitoring of herd immunity, the generation of risk prediction models, and is also likely to play a key role in the context of the global vaccination strategy [3,4]. Anti-SARS-CoV-2 neutralizing antibodies (NAbs) are of particular importance because these are the antibodies which inhibit the binding of the receptor-binding domain (RBD) of the surface spike (S) protein to the human angiotensin-converting enzyme 2 (ACE2) receptor. The complex formed between the virus S protein and the human ACE2 is responsible for the virus entry into hosts cells and the inhibition of the formation of this complex may thereby prevent infection and reduce disease severity [5,6].

Compared with SARS-CoV-2 antibody assays, which measured all the antibodies that are able to recognize the S protein, only assays measuring neutralizing antibodies (NAbs) reliably measure the real protective immunity of antibodies [7]. The current gold standard method to measure NAbs is the conventional virus neutralization test, which requires a biosafety level 3 laboratory to manipulate the live pathogen. These tests are reserved for very specialized laboratories and further require a high workload, skillful operators, and expensive installations, and they have a low throughput [8,9]. The use of a SARS-CoV-2 surrogate virus neutralization test (sVNT) based on antibody-mediated blockage of the interaction between the ACE2 receptor protein and the RBD has been found to be an attractive alternative [8,10,11].

In this study, we investigated neutralizing capacity by means of an sVNT in (1) previous COVID-19 patients and (2) volunteers vaccinated with BNT162b2. The specificity of the sVNT and its agreement with six SARS-CoV-2 antibody tests were also determined. A subset of samples was also tested with a pseudovirus neutralization test (pVNT).

## 2. Materials and Methods

### 2.1. COVID-19 Patients and Vaccinated Recipients

Demographic data for the two groups are presented in Table 1. In the COVID-19 patient group, 150 samples from 75 patients with a confirmed SARS-CoV-2 RT-PCR were retrospectively included from 26 March 2020 to 6 January 2021. Among them, 39 were females (median age = 45; min–max: 24–95 years) and 36 were males (median age = 62; min–max: 24–88 years). Multiple sequential sera were available for 41 patients. Seventeen patients required hospitalization and were categorized as moderate–severe patients, according to the WHO categorization [12]. Information on the days since the onset of symptoms was collected from medical records and was available for 63 patients. When data about symptoms were not available (*n* = 12), the day of diagnosis (i.e., the RT-PCR result) was used instead. The median time since diagnosis was 169 days (range, 11–296) and 139 days (range, 10–290) in mild and moderate–severe COVID-19 patients, respectively (*p*-value = 0.39).

In the second group, 90 healthcare volunteers who were scheduled to receive the complete dose regimen of the BNT162b2 mRNA COVID-19 vaccine were prospectively enrolled. Among them, 71.1% (*n* = 64) were females (median age = 44 years; range, 25–64 years) and 28.9% (*n* = 26) were males (median age = 48 years; range, 25–63 years). Thirty persons had a previous positive RT-PCR diagnosis (median days since RT-PCR = 158; range, 46–337). Among these, 29 persons were classified as mild cases and had positive anti-NCP antibodies, whereas only one was asymptomatic (positive RT-PCR diagnosis and no anti-NCP antibodies detected). Participants received the first vaccine dose from 25 January 2021 to 16 February 2021. The second dose was administered 21 days after the first one. All volunteers had blood drawn within 2 days before the first vaccine dose and additional blood samples were then collected after 14, 21, 28, 42, and 56 days.

Additionally, 250 samples collected before January 2020 were assessed to evaluate the clinical specificity of the sVNT.

### 2.2. Sample Collection

Blood samples were collected in serum-gel tubes (BD SST II Advance^®^, Becton Dickinson, NJ, USA) and centrifuged for 10 min at 1740× *g* on a Sigma 3-16KL centrifuge. Sera were stored in the laboratory serum biobank at −20 °C from the collection date. Frozen samples were thawed for 1 h at room temperature on the day of the analysis. Re-thawed samples were vortexed before the analysis. All samples were collected at the Clinique Saint-Luc (Bouge, Namur, Belgium). The study protocol was in accordance with the Declaration of Helsinki. All vaccinated participants provided informed consent prior to the collection of data and specimens (EudraCT registration number: 2020-006149-21).

### 2.3. Analytical Procedures

Neutralizing capacity was estimated by performing an sVNT. The iFlash-2019-nCoV NAbs assay is a one-step competitive paramagnetic particle chemiluminescent immunoassay (CLIA) for the quantitative determination of 2019-nCoV NAbs in human serum and plasma. The assay detects NAbs that block the binding of RBD and ACE2. First, NAbs (if present) react with the RBD antigen coated on paramagnetic microparticles to form a complex. Second, the acridinium-ester-labeled ACE2 conjugate is added to competitively bind to the RBD-coated particles, which have not been neutralized by the NAbs (if present) from the sample, and these form another reaction mixture. Under a magnetic field, magnetic particles are adsorbed to the wall of the reaction tube, and unbound materials are washed away by the wash buffer. The resulting chemiluminescent reaction is measured in relative light units (RLUs), with an inverse relationship between the amount of NAbs and the RLU value detected. According to the manufacturer, it shows excellent positive (98.5%) and negative percentage agreement (96.1%) with the conventional plate reduction neutralization test (PRNT). A result <10.0 AU/mL is considered negative and a result ≥10.0 AU/mL is considered positive (according to the manufacturer’s information). The sVNTs were performed on an iFlash1800 automated magnetic CLIA (MCLIA) analyzer from Shenzhen YHLO Biotech Co., Ltd. (Shenzhen, China) Internal quality controls (negative and positive) and 6 sera from COVID-19 patients at various NAbs titers were analyzed 10 times in a row to calculate the within-run precision of the assay. The positive internal quality control was also analyzed for a period of 15 days to calculate the between-run precision.

A total of 71 random samples (i.e., 23 pre-pandemic and 48 past-COVID-19 patient samples) were also assessed by means of a pVNT. Details about the method are presented in Appendix A. A sample is considered negative if the half maximal inhibitory concentration (IC50) value of this sample is below the dilution 1/20.

All samples from the first group, which was composed of COVID-19 patients, were also analyzed on 6 commercial immunoassays, namely: the Roche nucleocapsid (NCP) total antibody assay (positivity cut-off = 1.0 cut-off index (COI)), the Roche RBD total antibody assay (positivity cut-off = 0.8 U/mL), the DiaSorin S1/S2 IgG assay (positivity cut-off = 15 AU/mL), the Ortho S1 IgG assay (positivity cut-off = 1.0 S/V (sample signal/threshold value)), the Ortho S1 total antibody assay (positivity cut-off = 1.0 S/V), and the Phadia S1 IgG assay (positivity cut-off = 10 U/L), as described elsewhere [13]. The Roche NCP total assay was also used to determine the serological status of vaccinated participants before vaccine injection.

In group 1 and in previously infected individuals from group 2, RT-PCR for SARS-CoV-2 determination in nasopharyngeal swab samples was performed on the LightCycler^®^ 480 Instrument II (Roche Diagnostics^®^ (Rotkreuz, Switzerland)) using the LightMix^®^ Modular SARS-CoV *E*-gene set.

### 2.4. Statistical Analysis

Descriptive statistics were used to analyze the data. Sensitivity was defined as the proportion of correctly identified COVID-19 positive patients who were initially positive, according to an RT-PCR SARS-CoV-2 determination in nasopharyngeal swab samples. Specificity was defined as the proportion of pre-pandemic samples classified as negative. A Mann–Whitney test was used to assess potential differences in median time since diagnosis in mild versus moderate–severe COVID-19 patients. A linear regression model was implemented to evaluate the long-term kinetics of NAbs in past-COVID-19 patients. A simple linear regression and Pearson correlations were computed to assess the potential association between NAb titers and antibody titers obtained using 6 non-neutralizing commercial methods. Inter-rater agreements were also determined. The coefficients of variation (CV) ((standard deviation/mean) × 100 (%)) of the quantitative titers were used to determine the repeatability and intermediate precision of the assay. NAb titers among the two vaccinated groups at different time points were tested using an ANOVA multiple comparisons test. A *p*-value < 0.05 was used as a significance level. Data analysis was performed using GraphPad Prism^®^ software (version 9.1.0, San Diego, CA, USA).

## 3. Results

### 3.1. Clinical Specificity and Precision of the Assay

Considering the cohort of 250 pre-pandemic samples, only one sample was above the positivity threshold of 10 AU/mL (i.e., 15.7 AU/mL), leading to a specificity of 99.6% (CI 95%: 97.8–99.9%). The mean of the NAb titers was 3.0 AU/mL (CI 95%: 2.7–3.2 AU/mL) (Figure 1). The within-run CV ranged from 4.1% to 15.0% for NAb titers, ranging from 11.2 to 802.2 AU/mL. A higher CV was observed using the negative quality control (45.6% at a concentration of 4.4 AU/mL (min-max, 1.8–8.8 AU/mL) (Table 2). The between-run CV using the positive internal quality control was 10.0%.

### 3.2. sVNT vs. pVNT

Over the 71 samples tested in pVNT and sVNT, the agreement between the two methods was 97.2%. One sample was considered positive by pVNT but negative by sVNT, and one sample was considered negative by pVNT but positive by sVNT. These were the only two discordant results out of 71 samples, and they were close to the positivity cut-off of the sVNT (i.e., 9.6 and 10.1 AU/mL, respectively).

### 3.3. Neutralizing Antibodies in COVID-19 Patients

Figure 1 represents the NAb titers obtained in past-COVID-19 patients. The mean NAb titer in moderate–severe patients was significantly higher compared to mild patients (125 versus 33.9 AU/mL, *p*-value = 0.0006). All moderate–severe patients had positive NAbs (39/39) and 80.2% of mild patients were positive (89/111).

Considering only samples obtained ≥14 days since diagnosis, a weak but significant decay in NAb titers was observed over time in moderate–severe COVID-19 patients (r = −0.35, *p*-value = 0.03). The apparent slow decrease observed in mild COVID-19 patients was not statistically significant (r = −0.14, *p*-value = 0.14) (Figure 2).

The correlations between NAbs and SARS-CoV-2 antibody assays is presented in Figure 3. The six assays were significantly correlated to NAbs (*p*-value < 0.0001). The highest correlation coefficient was observed with the Phadia S IgG assay (r = 0.89) and the lowest one was observed on the Roche NCP assay (r = 0.46). With the exception of the Roche S total and Ortho IgG assays, higher correlations were obtained for IgG assays and weaker correlations for total assays (Figure 3). The agreement between methods was good and ranged from 82.7% to 88.0%.

### 3.4. Neutralizing Antibodies in Vaccinated Volunteers

The Figure 4 represents the evolution of NAbs in a group of 90 vaccinated individuals. In uninfected, seronegative individuals (*n* = 60/90), none had detectable anti-NCP antibodies nor NAbs at baseline. At day 14, the rate of seroconversion after the first dose was 83.3% (*n* = 50/60) with a 5.1-fold increase of NAb titers. Seven days after the administration of the second dose, a 114.3-fold increase was observed from baseline and all individuals had NAb titers above the positivity threshold. At days 42 and 56, the mean titers were not statistically different from those obtained at day 28 (*p*-value > 0.99) (Figure 4).

In individuals with a previous SARS-CoV-2 infection, 26.7% (*n* = 8/30) had negative NAbs at baseline and all individuals had positive anti-NCP results. At day 14, a significant 31.3-fold increase in NAbs was observed, with all individuals becoming positive. Compared to the NAb titers observed at day 14, the second dose administration had no significant impact on the NAb titers until up to day 56 (*p*-value > 0.99) (1.3-fold increase) (Figure 4).

Considering each time point separately, NAbs were always statistically higher in previously infected individuals compared to uninfected individuals (Figure 4). The mean NAb titers of previously infected individuals at baseline were not different from those observed in uninfected individuals 14 days after the first dose administration (*p*-value > 0.99). NAbs titers in previously infected individuals at day 14 were not different from titers obtained in uninfected individuals at days 28 and 56 (*p*-value > 0.05).

All vaccinated participants had significantly higher NAb titers after the complete dose regimen of the BNT162b2 vaccine compared to our cohort of COVID-19 patients (Figure 5).

## 4. Discussion

In this study, we evaluated the neutralizing capacity in two groups of COVID-19 patients and healthcare professionals who had received the complete dose regimen of the BNT162b2 vaccine. For that purpose, an sVNT was used. The method was based on antibody-mediated blockage of the interaction between the ACE2 receptor protein and the RBD. Since some reports demonstrated that some non-RBD targeting antibodies could possess neutralizing capacity [14,15], the agreement of the sVNT with pVNT was evaluated using a subset of our cohort of COVID-19 patients. An excellent agreement of 97.2% was found and is consistent with the manufacturer’s data. We also found that the specificity of the sVNT using a panel of 250 pre-pandemic samples was excellent (i.e., 99.6%) using the manufacturer’s cut-off of 10.0 AU/mL. A potential cut-off refinement using a ROC curve analysis did not reveal the usefulness of an optimized cut-off, as already performed for some serological assays [16,17,18,19,20,21,22]. The excellent specificity observed in our study was in line with that claimed by the manufacturer (i.e., 99.3%) using 270 samples from healthy volunteers who had no COVID-19 infection history and no vaccination history (manufacturer’s information). The precision of the assay was also good (Table 2).

As observed in previous reports [23], a stronger neutralizing activity was identified in moderate–severe compared to mild COVID-19 patients (Figure 1).

The slow decay in NAbs with time was also consistent with some reports [23,24,25,26,27,28,29,30], especially considering mild–moderate patients. A stronger SARS-CoV-2 antibody response in severe patients was also reported [13]. Compared to SARS-CoV-2 antibody assays, only neutralization activity assays reliably measure the real protective immunity of generated antibodies. There is also a high demand for the neutralization tests in specific clinical and industrial settings (e.g., for identification purposes with convalescent plasma or to support the development of vaccines). However, the conventional virus neutralization test requires live pathogens and is reserved for very specialized laboratories, requiring a high workload, skillful operators, specific and expensive facilities, and a biosafety level 3 laboratory, and on top of that, they have a low result throughput [8,9]. The use of automated and quantitative assays with a short turn-around time that have a well-documented correlation with the neutralizing activity should be preferred [7,9,31]. In our study, we observed that the Phadia S1 IgG assay had the highest correlation compared to sVNT (r = 0.89) (Figure 3). The second, better correlated assay was the DiaSorin S1/S2 assay (r = 0.75). This is in line with the findings of Legros et al., who showed a correlation of 0.71 using a microneutralization assay [32]. The Ortho S1 IgG assay had a higher correlation compared to the Ortho S1 total assay, as observed in a study by Padoan et al. [7]. Considering anti-NCP antibodies, the Roche total assay presented the lowest correlation with the results of the sVNT (r = 0.46). Patel et al. obtained similar conclusions when comparing the Roche NCP total assay to neutralizing activity (r = 0.40) [33]. We therefore confirm that the strongest correlations are observed using anti-S or anti-RBD assays [5,29,34,35,36] and our study highlights that correlations were especially high with the IgG assay. The fact that anti-NCP assays had a low correlation with the neutralizing activity was expected, as NAbs are directed against the S protein [37]. Nevertheless, it is important to keep in mind that a few patients may develop specific antibodies, i.e., antibodies detected by conventional serological assays, which do not translate into a detectable neutralizing activity. We therefore think that the assessment of the neutralizing activity using an sVNT on an automated platform (without the disagreement of the gold standard technique) might be valuable.

In the group of vaccinated individuals from the CRO-VAX HCP study [38,39,40], we evaluated the neutralizing response in a cohort of 90 volunteers, of which 60 were uninfected and 30 were previously infected by SARS-CoV-2, having received the complete dose regimen of the BNT162b2 vaccine. NAbs were measured at baseline, i.e., just before the administration of the first dose, and at 14, 28, 42 and 56 days. So far, few reports have investigated the neutralizing response in vaccinated subjects [41,42,43,44,45,46] and they mainly included few participants, only investigating the effect of the first dose [42,43,45], or did not include previously infected individuals [46]. In our study, a significant increase in NAb titers was seen after the first dose (i.e., a 5.1- and a 31.1-fold increase in uninfected and previously infected individuals, respectively) in all participants (Figure 4). Interestingly, the neutralizing capacity was similar when comparing previously infected individuals at baseline and naive individuals after the first dose, an observation that is similar to that of Manisty et al. using the Roche RBD total assay [47]. After the second dose, a significant increase in NAb titers was only observed in uninfected individuals (i.e., a 22.3-fold increase between day 14 and 28). Afterwards, the peak of the neutralizing capacity seems to have been reached at day 42 (i.e., 613 AU/mL) and a slight but non-significant decrease was observed at day 56 (527 AU/mL), which could be explained by the natural clearance of antibodies via excretion or mostly via catabolism [48]. Terpos et al. obtained similar findings using the cPass™ sVNT from GenScript [46]. All participants were considered positive 7 days after the second dose. In previously infected individuals, NAb titers at days 28 to 56, i.e., 7 and 35 days after the second dose, were not significantly different from those at day 14 after the first dose (Figure 4). The non-significant differences between the neutralizing capacity after the first dose and after the second dose support the concept only one dose might be sufficient to generate a complete neutralizing antibody response in individuals with a previous SARS-CoV-2 infection (Figure 4). Using an sVNT, Ebiger et al. also noticed a similar response after the second dose in previously infected individuals, but the number of participants who had received the second dose was low (*n* = 11) and they were followed up for a maximum of 28–42 days [41]. Evaluation of the pre-vaccinal serological status could therefore be proposed as a strategy to identify patients who will only require the booster dose [47]. In this context, pan-immunoglobulin assays should be preferred due to their higher sensitivity observed in long-term studies (up to 1 year post-infection) [13,49] compared to Nabs, which were negative in eight out of 30 (73.3%) previously infected individuals in our cohort (median days since RT-PCR = 158) (Figure 4). The NAb titer after the first dose in previously infected individuals was not significantly different from the NAb titers of uninfected individuals after the two-dose regimen (*p*-value > 0.05), even if lower mean titers were reported (Figure 4). This finding is inconsistent with the recent data of Anchini et al., who reported significantly higher NAb titers in previously infected individuals after the first dose compared to the uninfected individuals who had received two doses [44].

Our study (EudraCT registration number: 2020-006149-21) has a planned follow-up of two years. We will therefore be able to determine the long-term kinetics of the humoral response in both uninfected and previously infected participants.

In conclusion, we found a stronger neutralizing capacity in moderate–severe versus mild COVID-19 patients, in which a slow decay with time was observed. Vaccinated participants had significantly higher NAb titers after the complete dose regimen of the BNT162b2 vaccine compared to our cohort of COVID-19 patients. In light of these data, we can hypothesize that only one dose of the BNT162b2 vaccine might be sufficient in previously infected individuals to generate sufficient NAb titers to confer a sufficient serological immunity.

## Figures and Tables

**Figure 1 viruses-13-01364-f001:**
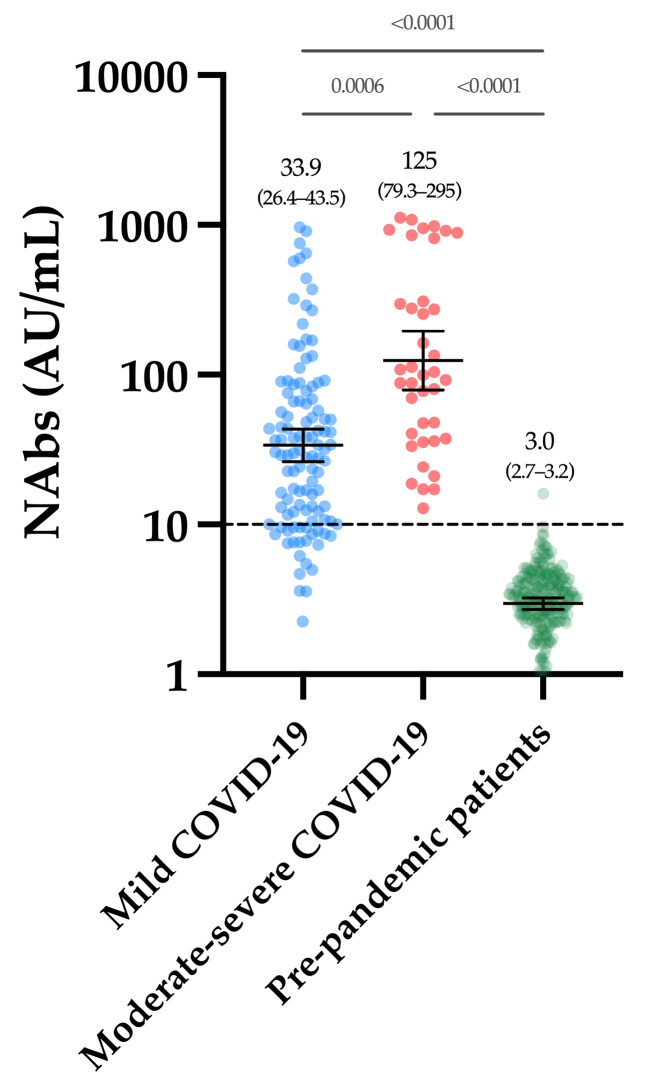
NAb titers obtained in the first group of COVID-19 patients and in the pre-pandemic cohort. The black dotted line corresponds to the positivity threshold of 10 AU/mL.

**Figure 2 viruses-13-01364-f002:**
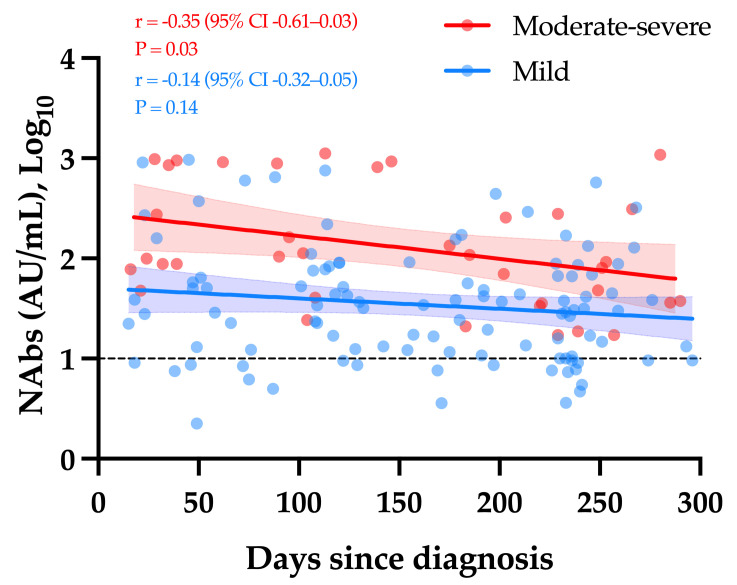
The kinetics of NAbs in moderate–severe versus mild COVID-19 (group 1). The black dotted line corresponds to the positivity threshold of 10 AU/mL.

**Figure 3 viruses-13-01364-f003:**
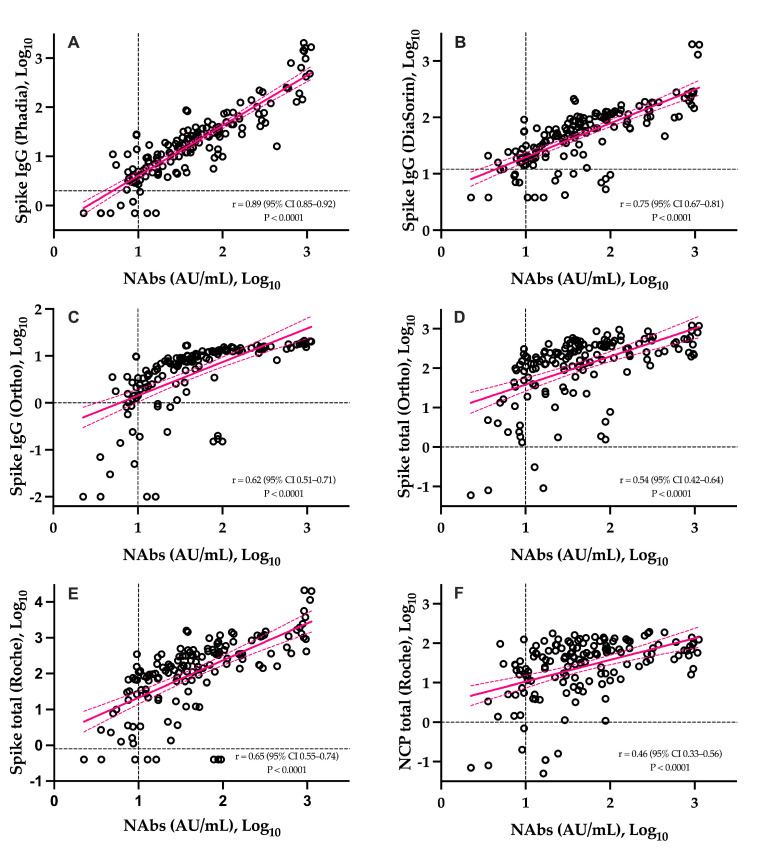
Head-to-head comparison of the sVNT to six different SARS-CoV-2 antibody assays. Black dotted lines correspond to the positivity threshold of each assay. (**A**): Phadia IgG spike assay; (**B**): DiaSorin IgG spike assay; (**C**): Ortho IgG spike assay; (**D**): Ortho total antibody spike assay; (**E**): Roche total antibody spike assay; (**F**): Roche total antibody nucleocapsid assay.

**Figure 4 viruses-13-01364-f004:**
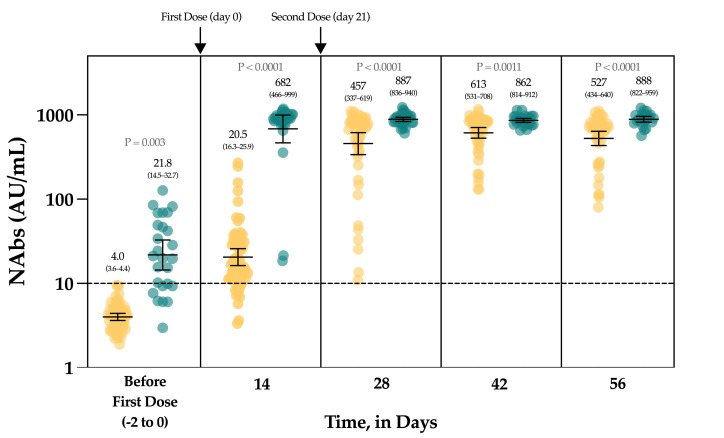
The evolution of NAbs in a group of 90 vaccinated participants. Uninfected individuals are represented in yellow and previously infected individuals are represented in green turquoise. The black dotted line corresponds to the positivity threshold of 10 AU/mL.

**Figure 5 viruses-13-01364-f005:**
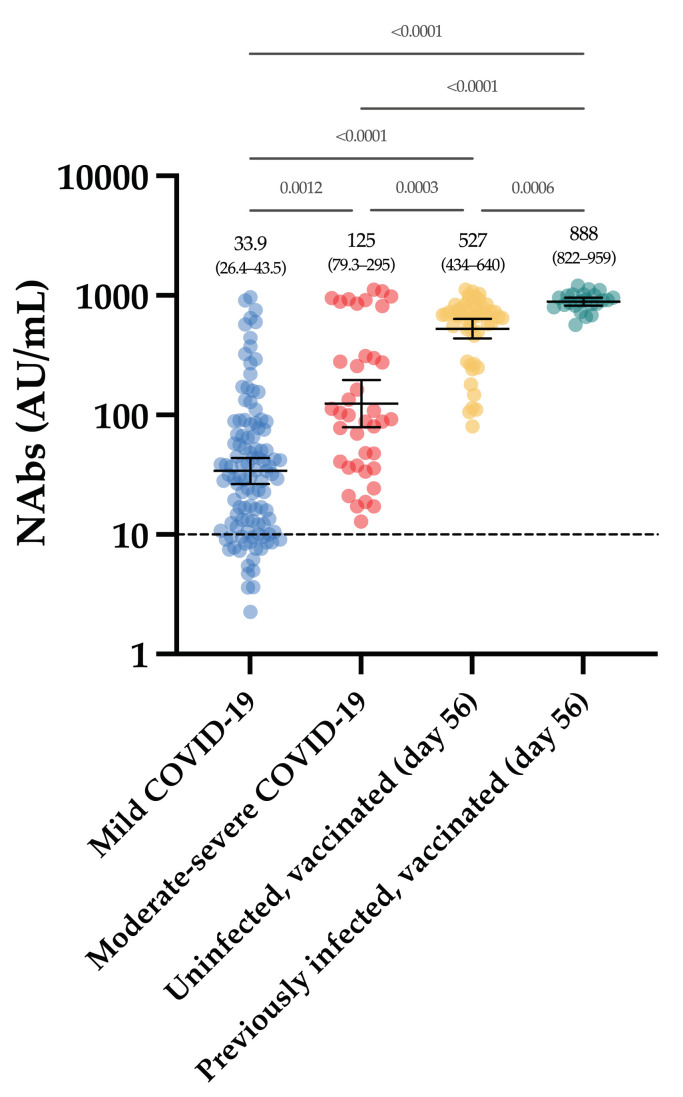
NAb titers obtained in the first group (moderate–severe and mild COVID-19), compared to those obtained in the group of vaccinated participants, at day 56. The black dotted line corresponds to the positivity threshold of 10 AU/mL.

**Table 1 viruses-13-01364-t001:** Demographic data for (1) the past-COVID-19 group and (2) the vaccinated group. The difference between the total number of samples and the number of patients/subjects is explained by multiple timepoints for blood sampling.

Demography	
*Group 1: Previous COVID-19 patients (n)*	*75*
**Females (*n* (%))**	**39 (52%)**
Age (median (min–max))	45 (21–95)
**Males (*n* (%))**	**36 (48%)**
Age (mean (min–max))	62 (24–88)
**Moderate–severe (*n* (%))**	**17 (22.7%)**
Time since diagnosis (median, (range))	169 (11–266)
**Mild (*n* (%))**	**58 (77.3%)**
Time since diagnosis (median, (range))	139 (10–290)
**Total number of samples**	**150**
***Group 2: BNT162b2 vaccine recipients (n)***	***90***
**Females (*n* (%))**	**64 (71.1%)**
Age (mean (min–max))	44 (25–64)
**Males (*n* (%))**	**26 (28.9%)**
Age (mean (min–max))	48 (25–63)
**Patients with a previous RT-PCR + (*n* (%))**	**30 (33.3%)**
Time since diagnosis (median, (range))	158 (46–337)
Moderate–severe (*n* (%))	0 (0.0%)
Mild (*n* (%))	29 (96.7%)
Asymptomatic (*n* (%))	1 (3.3%)
**Total number of samples**	**550**

**Table 2 viruses-13-01364-t002:** Precision of the sVNT using controls and patient samples. All materials were analyzed 10 times in a row.

	Neg. Control	Pos. Control	Sample A	Sample B	Sample C	Sample D	Sample E	Sample F
YHLO NAb Assay	6.36	54.3	10.8	42.0	266.9	576.2	727.6	783.8
	3.85	50.7	10.6	42.2	263.3	634.9	799.8	827.1
	1.84	55.3	14.7	45.6	278.0	667.8	837.1	867.4
	4.02	51.8	11.3	40.4	286.6	856.4	854.9	789.7
	3.28	60.6	10.2	41.2	287.3	863.0	739.4	814.3
	2.33	52.5	9.89	39.3	280.2	832.4	726.9	770.7
	4.18	53.2	10.5	41.3	250.2	609.5	796.7	820.7
	8.83	53.2	11.1	43.8	269.7	827.6	787.4	785.5
	5.03	56.8	13.0	42.3	292.2	765.3	799.4	811.1
	4.43	53.0	9.81	42.2	271.0	823.8	753.6	751.5
**Mean**	**4.42**	**54.1**	**11.2**	**42.0**	**274.5**	**745.7**	**782.3**	**802.2**
**SD**	**2.01**	**2.83**	**1.54**	**1.74**	**12.8**	**111.7**	**44.5**	**32.9**
**CV (%)**	**45.6**	**5.23**	**13.8**	**4.14**	**4.67**	**15.0**	**5.69**	**4.11**

## Data Availability

The data presented in this study are available on request from the corresponding author. The data are not publicly available according to ethical committee decision on the conduct of this study.

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
