# Peer review of "Neutralizing Antibodies in COVID-19 Patients and Vaccine Recipients after Two Doses of BNT162b2"

_viruses, 2021, doi:10.3390/v13071364_

Round 1
Reviewer 1 Report
This is a very interesting study that used a surrogate virus neutralizing antibody assay to detect titers of RBD-binding antibodies in the sera of carefully selected individuals who received the BNT162b2 vaccine. The study is of interest to the readership of this journal and the biomedical and clinical science audience.
Minor Comments:
- Although one of the greatest strengths of this study, when compared to recent studies examining similar questions, is the inclusion of previously infected individuals as vaccine recipients. Interestingly, using an assay that detected antinucleocapsid antibodies, close to 30% of infected individuals who had antinuclocapsid antibodies did not have neutralizing antibodies. It would therefore have been very helpful to screen even the non-infected participants for antinucleocapsid antibodies in this study. How else could the authors have ascertained that the so-called uninfected individuals were not infected participants who were previously exposed but developed low levels of antibodies that are now undetectable? Since the authors aim to continue monitoring antibodies in the participants, it would be important to consider screening for antinucleocapsid antibodies in all participants, irrespective of history of active infection (detected using RT-PCR).
- The authors basically compared their findings to those of others. It would have been interesting to see their take on the possible causes of decline in neutralizing antibodies over time
Reviewer 2 Report
Title of the manuscript: Neutralizing Antibodies in COVID-19 Patients and Vaccine Recipients After Two Doses of BNT162b2.
Manuscript ID: viruses-1237233
Evaluation Summary: The present work is aimed to study pre-existing antibody status and antibody kinetics upon vaccination in healthcare workers being given two doses of BNT162b2. Data suggests that single dose is sufficient for subjects who were already seropositive. However, initially seronegative individuals vaccinated participants had significantly higher NAbs titers after the complete dose regimen of BNT162b2 vaccine compared to our cohort of COVID-19 patients. This paper is of interest to readers who are working on evaluating efficacy of BNT162b2 vaccine.
Favresse J et al., has assessed the effect of vaccination that involved BNT162b2 COVID 19 vaccine against SARS-CoV-2. A total of 165 individuals were recruited for this study by measuring antibody responses against spike(S) protein of SARS-CoV-2. First group was 75 patients with a confirmed 62 SARS-CoV-2 RT-PCR, second group was 90 healthcare volunteers having received who were planned to receive the complete dose regimen of the BNT162b2 mRNA COVID-19 vaccine were prospectively. Authors have determined neutralizing activity with these two cohorts, compared sVNT results to 6 SARS-CoV-2 antibody assays concluding that one dose of the vaccine provides immunity to previously infected subjects while uninfected individuals require booster dose to achieve neutralizing activity.
Strengths: Having access to cohorts of who are seropositive before vaccination and seronegative HCW’s who are naïve to infection.
Weakness: Not using appropriate tools to assess antibody titers and neutralizing capacity
Recommendations/Comments to authors:
Major:
- SARS-CoV-2 Neutralization Antibody Detection Kit used by authors is just a surrogate virus neutralization test (sVNT), a serological assay to determine the presence of RBD blocking antibodies that compete for human ACE2 binding but does not assess actual virus neutralization capacity. There are several manuscripts presenting non-RBD targeting antibodies possessing neutralizing capacity (please see below for references).
- Suryadevara, Naveenchandra, Swathi Shrihari, Pavlo Gilchuk, Laura A. VanBlargan, Elad Binshtein, Seth J. Zost, Rachel S. Nargi et al. "Neutralizing and protective human monoclonal antibodies recognizing the N-terminal domain of the SARS-CoV-2 spike protein." Cell (2021).
- Chi, Xiangyang, Renhong Yan, Jun Zhang, Guanying Zhang, Yuanyuan Zhang, Meng Hao, Zhe Zhang et al. "A neutralizing human antibody binds to the N-terminal domain of the Spike protein of SARS-CoV-2." Science 369, no. 6504 (2020): 650-655.
- Usually, sVNT demonstrated a high non-neutralizing antibody detection rate. This has been evaluated in case SARS-2 sVNT kits as well and found that agreement between sVNT and PRNT-50 was moderate. Hence, conclusions about neutralization capacity are contradictory by using sVNT .
Minor:
- Authors should explain how false negative rates from sVNT are handled, since false negative rates are proved to be higher using this kit.
- I recommend authors to perform at least pseudo neutralization assay for few samples and correlate that with sVNT assay results to check for consistency of results.
- For readers convenience, Authors should provide more details of how assays are performed in methods section instead of according to the manufacturer’s protocol at least in brief.

Reviewer 3 Report
The authors assessed the neutralizing capacity of anti-SARS-CoV-2 antibodies 75 COVID-19 patients and 90 vaccinated individuals by with a surrogate virus neutralization test. I believe the paper will be of interest to the readers and would recommend it for acceptance after the minor points listed below and annotated on the manuscript are addressed.
- Please covert the information section 2.1. to Table format o that easy to understand.
- Line 143: Please include number of samples.
- Supplemental Figure 1, Figure 1 and Figure 4: Please explain the meaning of horizontal bars.
- Figure 4: Please change colors.
- Line 231: Please change sVRT to sVNT.
- If possible, please include WHO standard sera data.
Reviewer 4 Report
The authors compared the neutralizing antibodies responses in COVID-19 patients and in vaccinated individuals using a surrogate virus neutralization test on an automated platform. The authors performed a proper data analysis and gave a good discussion of the data. The results and conclusion are overall consistent with previous reports. In addition, they provided evidence for a slow decay of neutralizing antibodies in COVID-19 patients up to 300 days post-diagnosis and confirmed other reports that one dose of vaccine should be enough in exposed individuals. However, the authors must fix the following minor issues.
- Please check typos or grammar in lines 68, 72, and 106.
- Suggest to include supplementary figure 1 into figure 1 as panel B and show values from each individual.
- Consider rephrasing the sentence in line 185.
- Please indicate the time points for NAbs detections in Figure 2.
- Line 194 is very confusing since the authors did not show correlation-related data in figure 1. Please show the data.
- Please indicate which cohort was analyzed in Figure 3.
- No color symbols were shown in Figure 4 as described. Please fix the problem.
Round 2
Reviewer 2 Report
Authors have addressed my concerns raised during first review and has incorporated new results which significantly enhanced quality of the manuscript. Hence, I recommend this manuscript for publication.